# Unique Metabolomic Profile of Skeletal Muscle in Chronic Limb Threatening Ischemia

**DOI:** 10.3390/jcm10030548

**Published:** 2021-02-02

**Authors:** Ram B. Khattri, Kyoungrae Kim, Trace Thome, Zachary R. Salyers, Kerri A. O’Malley, Scott A. Berceli, Salvatore T. Scali, Terence E. Ryan

**Affiliations:** 1Department of Applied Physiology & Kinesiology, University of Florida, Gainesville, FL 32611, USA; rbk11@ufl.edu (R.B.K.); kimk1@ufl.edu (K.K.); trthome@ufl.edu (T.T.); zachary.salyers@ufl.edu (Z.R.S.); 2Division of Vascular Surgery and Endovascular Therapy, University of Florida, Gainesville, FL 32610, USA; kerri.omalley@surgery.ufl.edu (K.A.O.); bercesa@surgery.ufl.edu (S.A.B.); Salvatore.Scali@surgery.ufl.edu (S.T.S.); 3Malcom Randall Veteran Affairs Medical Center, Gainesville, FL 32608, USA; 4Center for Exercise Science, University of Florida, Gainesville, FL 32611, USA; 5Myology Institute, University of Florida, Gainesville, FL 32610, USA

**Keywords:** peripheral artery disease, peripheral vascular disease, hypoxia, metabolomics, metabolites, myosteatosis

## Abstract

Chronic limb threatening ischemia (CLTI) is the most severe manifestation of peripheral atherosclerosis. Patients with CLTI have poor muscle quality and function and are at high risk for limb amputation and death. The objective of this study was to interrogate the metabolome of limb muscle from CLTI patients. To accomplish this, a prospective cohort of CLTI patients undergoing either a surgical intervention (CLTI Pre-surgery) or limb amputation (CLTI Amputation), as well as non-peripheral arterial disease (non-PAD) controls were enrolled. Gastrocnemius muscle biopsy specimens were obtained and processed for nuclear magnetic resonance (NMR)-based metabolomics analyses using solution state NMR on extracted aqueous and organic phases and ^1^H high-resolution magic angle spinning (HR-MAS) on intact muscle specimens. CLTI Amputation specimens displayed classical features of ischemic/hypoxic metabolism including accumulation of succinate, fumarate, lactate, alanine, and a significant decrease in the pyruvate/lactate ratio. CLTI Amputation muscle also featured aberrant amino acid metabolism marked by elevated branched chain amino acids. Finally, both Pre-surgery and Amputation CLTI muscles exhibited pronounced accumulation of lipids, suggesting the presence of myosteatosis, including cholesterol, triglycerides, and saturated fatty acids. Taken together, these metabolite differences add to a growing body of literature that have characterized profound metabolic disturbance’s in the failing ischemic limb of CLTI patients.

## 1. Introduction

Chronic limb threatening ischemia (CLTI) is the most severe manifestation of peripheral arterial disease (PAD) and is clinically characterized by pain at rest, non-healing wounds, gangrene, and a high risk of limb amputation or death. These symptoms result in devastating mobility impairment and loss of independence. Current treatment approaches for CLTI include endovascular (atherectomy, angioplasty and/or stenting) and open revascularization (bypass grafting) procedures that directly attempt to restore arterial perfusion. A significant number of CLTI patients that undergo surgical intervention will still require limb amputation despite graft patency and/or improvements in arterial flow by angiography [1,2,3,4]. The reasons for high failure rates despite technically good restoration of limb blood flow is not fully understood. A critical barrier to developing more effect treatments is a lack of understanding of the resident tissue biology and its contribution to the overall pathobiology of CLTI.

Despite an important clinical focus on reestablishing limb blood flow, a strong predictor of morbidity and mortality in PAD patients is muscle function/exercise capacity [5,6,7,8,9,10,11]. Previous reports have highlighted pathological evidence of skeletal muscle myopathies and necrosis in PAD patients [12,13]. In fact, a recent study specifically reported that fragile muscle mitochondrial function distinguished mild PAD from CLTI patients [14]. Patients with PAD have been reported to suffer from decreased muscle metabolism and mitochondrial respiration, altered expression of mitochondrial enzymes, increased oxidative stress, and mutations in the mitochondrial genome [15,16,17,18,19,20,21,22,23,24,25,26,27,28]. Some of these metabolic deficiencies are present in both the skeletal muscle and muscle stem cells (satellite cells) of CLTI patients [14,29,30,31,32]. Moreover, it was recently reported that some aspects of the CLTI muscle mitochondrial phenotype may improve following successful vascular intervention [33], suggesting that muscle metabolism changes may be associated with clinical improvement. These exciting reports advocate for the possibility that therapeutic targeting of limb muscle metabolism in CLTI, which has shown early promise in preclinical studies demonstrating improved limb blood flow recovery and reduced tissue necrosis [34,35,36].

Metabolomics is an emerging field that facilitates characterization of metabolites in biological specimens including fluids, cells, tissues and/or whole organisms [37]. Nuclear magnetic resonance (NMR) and mass spectrometry (MS) are the two most commonly used analytical techniques in metabolomics. A single measurement from both of these techniques can provide important information about the molecular/chemical structure of metabolites. Both can be used to elucidate the chemical structure and determine the concentration of metabolites; however, both technologies have some limitations. MS is far superior in terms of sensitivity, with some metabolites being detected in the pico-molar range using small (microliter) sample volumes [38]. However, reproducibility is a major issue associated with MS, and batch-to-batch and instrument variability can reduce confidence. Another limitation of MS is its destructive nature, which requires extraction of metabolites, so MS cannot be used for in vivo and intact tissues-related studies. NMR, despite its lower sensitivity as compared to MS, is highly reproducible [39]. Moreover, NMR samples can be recovered for further analysis after acquiring NMR spectra. Real-time metabolic studies are possible with advanced NMR techniques such as dynamic nuclear polarization (DNP) [40]. Several in vivo studies are also possible with different variants of NMR [41]. Despite the differences in MS and NMR approaches, both allow for qualitative and quantitative metabolite determination with high throughput [42]; however, both techniques require a high level of technical expertise to properly run the equipment.

Two different variants of NMR had been used for metabolomics: (1) high-resolution magic angle spinning (HR-MAS), and (2) solution state NMR. HR-MAS NMR spectroscopy is an advanced technique that can be applied to the intact tissue samples ex vivo and most of the NMR pulses can be easily applied for this technique [43]. Spinning of tissue sample (~5 kHz) with an angle of 54.7° with respect to the B_o_ magnetic field helps to reduce the line broadening effects (that remains as major drawback in in vivo MRS) [43]. The sample preparation protocol is simple, and the samples can be re-used after HR-MAS NMR experiments for other biochemical studies such as histology and/or gene expression [44,45]. With HR-MAS NMR spectroscopy, acquisition of high-quality spectral resolution is possible (far better than in vivo MRS) [43]. This method is very useful for the samples that are not soluble in deuterated solvents. Despite having some advantages, it is limited to low-concentration metabolites and also gives slightly broad spectra. The solution state ^1^H NMR helps to overcome these issues of HR-MAS NMR spectroscopy. With solution state ^1^H NMR, quantitation of low-concentration metabolites is possible that is otherwise missed by HR-MAS NMR.

In this prospective study, a cross-sectional discovery analysis of non-PAD controls and CLTI patients undergoing either a vascular intervention or undergoing limb amputation was performed and involved a detailed assessment of the limb muscle metabolome using semi-solid and solution state nuclear magnetic resonance spectroscopy. It was hypothesized that patients undergoing limb amputation would present with altered muscle metabolite features compared with non-PAD controls.

## 2. Experimental Section

### 2.1. Study Populations

Gastrocnemius muscle specimens were obtained from ten older adult non-PAD controls (Control), ten patients with chronic limb threatening ischemia undergoing surgical intervention (CLTI Pre-surgery), and ten CLTI patients undergoing limb amputation (CLTI Amputation). Five Pre-surgery patients underwent bypass interventions and five underwent endovascular procedures. Inclusion criteria for the control group included being aged 60 years or older and having an ankle-brachial index (ABI) greater than 0.9 with the absence of non-compressible vessels. CLTI patients were included based on their clinical diagnosis of CLTI and medical necessity of either a surgical intervention or limb amputation (due to non-salvageable limb). Patients with non-atherosclerotic occlusive disease (vasculitis, aneurysm, embolic disease, Buerger’s disease, and acute limb ischemia) were excluded. Because CLTI patients routinely present with numerous co-morbid conditions, we did not exclude patients based on risk factors or other conditions (i.e., hyperlipidemia, kidney disease, hypertension, diabetes). CLTI patients were recruited from the vascular clinic’s at UF Health and the Malcom Randall VA Medical Center in Gainesville, FL, USA. Non-PAD control patients were recruited from the local community. Sample collection in this study occurred between July 2018 and December 2019. This study was approved by the institutional review board at the University of Florida and carried out according to the Declaration of Helsinki. All participants were fully informed about the research and informed consent was obtained. Clinical and physical characteristics of patients are shown in Table 1.

### 2.2. Muscle Specimen Collection

Muscle specimens were collected within the confines of the operating rooms (CLTI patients) or via percutaneous muscle biopsy using sterile procedures previously described [14,46]. A portion of the muscle was quickly trimmed of fat/connective tissue and snap frozen in liquid nitrogen for metabolomics analysis.

### 2.3. Chemicals

Chemicals used in this study were purchased from different vendors and used without further purification. Ethylene diamine tetra acetic acid (EDTA), sodium azide (NaN_3_), sodium monobasic and dibasic phosphates were used to prepare phosphate buffer and were obtained from Millipore-Sigma (St Louis, MO, USA). Pyrazine, an internal standard used with organic solvents, was also obtained from Millipore-Sigma. Deuterated chloroform (CDCl_3_) and deuterium oxide (D_2_O) were purchased from Cambridge Isotope Laboratories (Andover, MA, USA). D_6_-4,4-dimethyl-4-silapentane-1-sulfonic acid (DSS-D_6_, 98%) was obtained from FUJIFILM Wako (Richmond, VA, USA).

### 2.4. Metabolite Extraction

Both polar and non-polar metabolites were extracted from the gastrocnemius muscle specimens using modified FOLCH extraction method [47]. In brief, wet weigh of the frozen tissues were determined and immediately homogenized in 1 mL of ice-cold methanol using a PowerLyzer 24 Homogenizer (QIAGEN Group, Hilden, Germany). All enzymatic activities are halted once the sample was homogenized in the methanol. Homogenization was followed by centrifugation (13.2K r.p.m., 4 °C, 30 min) and supernatant was transferred into a new glass vial consisting a mixture of 3 mL of ice-cold chloroform and methanol (2:1 *v*/*v*) ratio. The cold mixture was vortexed for several minutes and left in an ice bath 15 min to allow for phase separation. Next, 1 mL of ice-cold 0.9% saline was added to the mixture followed by vigorous mixing. The mixture was again left in the ice bath for 45 min for phase separation. The upper methanol/water layer was transferred to a new falcon tube. To the lower chloroform layer, 1 mL of ice-cold 0.9% saline was added and all steps were followed as mentioned above. Following a 45-min incubation, upper methanol/water layer was again transferred to the previous falcon tube and dried using a Labconco freeze drier (Labconco Corporation, MO, USA). The chloroform layer was dried under a stream of nitrogen gas. The dried samples (both aqueous and organic phases) were stored at −80 °C until resuspension for NMR experiments.

### 2.5. Sample Preparation and NMR Acquisition

All raw metabolomics data have been deposited to the Metabolomics Workbench (https://www.metabolomicsworkbench.org) under the following Study ID’s: ST001615, ST001616, ST001617.

Lyophilized aqueous phase samples were re-suspended in 50 µL of 50 mM phosphate buffer (pH = 7.2) consisting 2 mM of EDTA along with 0.2% NaN_3_ and 0.5 mM D6-DSS in 100% deuterated environment. Lyophilized organic phase sample were re-suspended in 80 µL of CDCl_3_ along with 10 mM of pyrazine. All samples were loaded in 1.7 mm NMR tube to acquire spectra.

All solution state NMR experiments were acquired with a Bruker (Bruker BioSpin Corporation, Billerica, MA) Avance Neo 600 MHz/54mm console with a 1.7 mm TCl CryoProbe. First slice of 1D nuclear Overhauser effect spectroscopy (noesypr1D) [48] pulse sequence with water pre-saturation during relaxation delay (d1) was used to collect 1D spectra for both aqueous and organic phase samples. Acquisition parameters were applied as described previously [49,50,51]: 128 scans (nt), 1 s recycle delay (d1), 4 s acquisition time (aq), 100 ms mixing time, and 7142.9 Hz spectral width (sw) using ^1^H 90° pulse width (pw) at room temperature (25 °C).

#### HR-MAS NMR on Intact Gastrocnemius Muscle Specimens

Semi-solid HR-MAS spectra were collected on intact human gastrocnemius muscle specimens. For this, a Bruker 800 MHz system equipped with 4 mm HR-MAS probe was used. Preparation of HR-MAS was performed using 3.2 mm inside diameter plastic insert, following the protocols described by Downes et al. [52]. The following acquisition parameters were used to collect 1D NOESY spectra (noesypr1D) with pre-saturation of water signal: 256 nt, 2 s d1, 2.04 s aq, 100 ms mixing time, and 8012.8 Hz sw using 90° pw at 4 °C. The sample was spun at 5 kHz speed maintaining 54.7° magic angle.

### 2.6. Data Processing and Analysis

Spectra were processed with MestReNova 14.1.2-25024 software (Mestrelab Research, S.L., Santago de Compostela, Spain). A zero filling of 64K was performed with line broadening of 0.22 Hz before Fourier Transformation followed by phase and base-line correction (Splines). Spectra from the aqueous phase samples were calibrated and normalized with DSS peak at 0.00 ppm. The spectra from the organic phase samples were calibrated with the chloroform peak at 7.26 ppm and normalized with pyrazine peak at 8.61 ppm. HR-MAS proton spectra were referenced with the alanine doublet at 1.46 ppm. Extraction of integrated areas for the selected metabolites were done from these well-referenced and/or normalized spectra. Wet weigh correction was performed on these data and further used for plotting Box and Whisker plots as well as input of raw data into metaboanalyst4.0 for analysis (https://www.metaboanalyst.ca/) [53]. For aqueous phase samples, concentration of metabolites was calculated with respect to DSS (internal reference) peak area. Box and Whisker plots were generated using GraphPad Prism (version 9.0.0 (121), GraphPad Software, San Diego, CA, USA, www.graphpad.com).

### 2.7. Metabolites Assignment

Assignment of the metabolites was done on the basis of 1D and 2D spectra collected for a particular sample. Appendix A shows the assignment of the metabolites in 1D NOESY spectra from aqueous phase, organic phase and intact gastrocnemius samples. A different set of 2D spectra (Appendix A) was collected for an aqueous phase sample (control 6a) using the standard Bruker library for the verification of the metabolites. Biological magnetic resonance bank (BMRB) [54] data and several studies [55,56,57] were also used for the verification of metabolites.

### 2.8. Statistical Analysis

Shapiro-Wilk test confirmed the metabolite abundance data were normally distributed. Analysis of variance (ANOVA) was performed with Metaboanalyst 4.0 using a false discovery rate (FDR) corrected data. Principal component analysis (PCA) and partial least square discriminant analysis (PLS-DA) were also conducted. The supervised PLS-DA findings were further validated by Q^2^ and permutation tests. Metabolites showing variable importance in projection (VIP) scores greater than 1 (from PLS-DA analysis) were outlined as distinct metabolites. Furthermore, one-way analysis of variance was performed using GraphPad Prism (version 9.0.0 (121), GraphPad Software, San Diego, CA, USA, www.graphpad.com) with *p* ≤ 0.05 considered as being statistically significant. The results are expressed as mean ± standard deviation (SD) in the tables, and box and whisker plots were generated to show all points as well as median and ranges (whiskers = min. and max.). Analysis of clinical and physical characteristics was performed with either ANOVA or Chi-squared testing.

## 3. Results

### 3.1. Patient Physical and Clinical Demographics

This was a prospective cohort study that examined the metabolomic profile of skeletal muscle from CLTI patients undergoing surgical intervention or amputation, as well as a cohort of non-PAD controls. Clinical and physical characteristics of patients are shown in Table 1. CLTI patients exhibited severe symptomology (Rutherford Classification 3–6), with high incidence of common PAD risk factors including hypertension, hyperlipidemia, coronary artery disease, and diabetes. Fifty percent (*n* = 5) of the CLTI Amputation patients had a previous vascular intervention

### 3.2. Metabolomic Analysis of Gastrocnemius Muscle

Both high-resolution HR-MAS and ^1^H NMR (solution state) NMR spectroscopy methods coupled with multivariate analysis were used in this study to cultivate a metabolome profile for CLTI limb muscle. Overall graphical image showing the comprehensive work flow applied in this study can be seen in Figure 1. Gastrocnemius muscle specimens were collected from ten older adult non-PAD controls (Con), ten CLTI Pre-surgery patients, and ten CLTI Amputation patients.

Because of limitations with specimen size, ^1^H HR-MAS NMR spectra were collected on 24 intact muscles (8 controls, 6 CLTI Pre-surgery and 10 CLTI Amputation). Similarly, FOLCH extraction was performed on another set of 25 gastrocnemius muscle specimens and solution state ^1^H NMR spectra were collected on both aqueous and organic phase samples. Furthermore, multivariate analysis was performed on these ^1^H HR-MAS and ^1^H solution NMR spectra’ extracted datasets separately and metabolomics profiling was created. Use of HR-MAS NMR on intact tissue made it possible for the measurement of lipids (including lipo-proteins) and small molecular weight metabolites simultaneously. Moreover, the inclusion of solution state ^1^H NMR on FOLCH extracted tissue samples enabled the measurement of metabolites that were missed or difficult to quantify by HR-MAS because of their lower concentrations and/or overlapping spectra. These analyses defined a larger number of metabolites (having sharp peaks) with high resolution for ^1^H NMR spectra of FOLCH extracted tissue samples as shown in Appendix A (aqueous phase) and Appendix A (organic phase) as compared to HR-MAS (Appendix A).

Variation in peak intensities for the different metabolites across the three different groups can be clearly seen in Figure 2 and Appendix A. Metabolites such as lactate, alanine, acetate, glutamine, creatine, taurine, adenosine triphosphate/adenosine monophosphate (ATP/AMP), histidine, and formate were varying in representative ^1^H NMR spectra (Figure 2A–C) from the three groups for aqueous phase samples. On the other hand, different classes of lipids (with lipo-proteins) along with creatine, alanine, lactate, taurine, ATP/AMP, and histidine were clearly difference in ^1^H HR-MAS NMR spectra (Figure 3D–F). Slight variation in different lipid classes among the groups can be observed for organic phase samples too (Appendix A).

Next, we performed unsupervised PCA as well as supervised PLS-DA analyses of the wet weight normalized ^1^H NMR and HR-MAS datasets, as shown in Figure 3. These analyses revealed greater within-group variability for CLTI Amputation and clear clustering can be observed for CLTI Amputation specimens compared to CLTI Pre-surgery and non-PAD control muscles. Clustering was clearer in aqueous phase samples (Figure 3A (PCA) and 3D (PLS-DA)) and intact tissues HR-MAS samples (Figure 3B (PCA) and 3E (PLS-DA)) compared to organic phase samples ((Figure 3C (PCA) and 3F (PLS-DA)). In ^1^H HR-MAS NMR-based PCA score plots (Figure 3B), CLTI Pre-surgery and control were found to be overlapping in most cases. PCA components 1 and 2 comprise about 55% of the variables for aqueous phase extracted data (Figure 3A) and 50% of the variables for ^1^H HRMAS dataset. On the other hand, all three groups in PCA score plots of organic phase ^1^H NMR dataset (Figure 3C) exhibited some clustering, but overlapped with each other. The supervised PLS-DA approach slightly increased the clustering between CLTI Amputation with other two groups in ^1^H NMR (aqueous phase, Figure 3D) and ^1^H HR-MAS (Figure 3E) datasets, and again the CLTI Amputation group showed greater within-group variability. Both permutation test and Q^2^-value validate the PLS-DA approach for aqueous phase and HR-MAS samples, but not the organic phase samples. PLS-DA components 1 and 2 demonstrated about 45% of the total variance for ^1^H NMR (aqueous phase, Figure 3D) and about 67% for ^1^H HRMAS datasets (Figure 3E). However, again the ^1^H NMR dataset for organic phase samples was unable to produce strong separation among the three groups (Figure 3F).

Metabolites responsible for driving the separation of ^1^H-NMR metabolomics profiles in PLS-DA analysis for aqueous phase (^1^H NMR: solution state) and ^1^H HR-MAS datasets are shown in Table 2. In Table 2, metabolites/compounds with VIP scores greater than 1 are shown. Lactate was found to be common in driving separation between the three groups in PLS-DA analysis in both ^1^H NMR (aqueous phase) and ^1^H HR-MAS datasets with VIP score ≥ 1.5. Small metabolites such as glutamate, pyruvate, glycine, succinate, inosine, O-Phosphoethanolamine, creatinine, creatine phosphate, malonate, tyrosine, 3-methyl histidine, phyenylalanine and 2-Aminoadipate were found to be responsible for driving separation for the ^1^H NMR aqueous phase dataset. However, mostly, different classes of lipids (along with lipoproteins) were responsible in ^1^H HR-MAS dataset for driving separation among the groups. For organic phase samples (^1^H NMR: solution state), please refer to Appendix A.

### 3.3. CLTI Display Biomarkers of Ischemic Metabolism at Amputation but Not Prior to Surgery

NMR is well known for its quantitative properties and high reproducibility [50]. Intensity, as well as integral area of the peak of a particular metabolite in NMR spectrum, is the direct representative of its concentration [58]. The integral peak area of a representative peak of a particular metabolite was taken for quantitative purposes. The integral area of internal reference (0.5 mM DSS for aqueous phase samples) was used to determined concentration for few metabolites. Differences in the peak integral areas or calculated concentrations were compared among the three groups. Several unique metabolite changes were observed in non-salvageable CLTI limbs being amputated. First, these muscle specimens displayed classic biomarkers of hypoxic/ischemic tissues. For example, CLTI Amputation specimens had significantly higher concentrations of lactate, succinate, fumarate, alanine, as well as a marked decrease in the pyruvate/lactate ratio (Figure 4). These metabolite changes were consistent with observations reported in the ischemic myocardium and is indicative of metabolic changes necessary to support substrate level (non-oxidative) phosphorylation [59].

### 3.4. Dysregulated Amino Acid Metabolisms Are Distinguising Charactertistics of CLTI

Another divergent characteristic of non-salvageable CLTI limb muscle was altered amino acid metabolism. Specifically, CLTI Amputation muscles displayed accumulation of branched-chain amino acids (BCAAs)—isoleucine, leucine, and valine—as well as alanine, phenylalanine, tyrosine, and glycine. These findings imply an imbalance between protein synthesis and degradation (Figure 5). Interestingly, all of these amino acid changes were absent in CLTI Pre-surgery muscle specimens, suggesting that dysregulated amino acid metabolism may be a “treatable” target to improve limb salvage in CLTI. Notably, 3-methylhistidine, a biomarker of muscle protein degradation, was elevated in CLTI Pre-surgery specimens. 

### 3.5. Lipidomic Differences in CLTI Muscle Are Indicative of Myosteatosis

Using HR-MAS and extracted organic phase samples, we also detected altered lipid profiles in CLTI muscle specimens. Unlike the majority of aqueous phase metabolites, the lipid species were similar between CLTI Pre-surgery and Amputation specimens and were generally more abundant when compared to non-PAD control muscles (Figure 6). This observation suggests that CLTI patients suffer from the accumulation of ectopic fat within muscle, clinically termed myosteatosis. Key detected lipid differences include elevated cholesterols (Figure 6A), triglycerides (Figure 6B), and saturated fatty acids (Figure 6C). A mixture of unsaturated fatty acids also showed similar trend as saturated fatty acids and showed significant increase in CLTI patients compared to the control group (Appendix A). Peaks from methylene protons associated with double bonds (chemical shift range of 1.98–2.07 ppm) and divinyl methylene protons of unsaturated fatty acids (chemical shift range of 2.74–2.87 ppm) were significantly elevated in CLTI patients. While NMR is limited in its ability to distinguish individual fatty acids in mixture, these results are clearly indicative of elevated levels of both saturated and unsaturated fatty acids withing CLTI muscles.

## 4. Discussion

Chronic limb threatening ischemia is the most severe form of PAD and carries a high risk for amputation and mortality. Generally speaking, CLTI patients present with more complex patterns of atherosclerosis and in most cases surgical intervention is often performed with the primary goal of limb salvage. Unfortunately, limb amputation rates remain high, despite technically good endovascular and revascularization procedures. Skeletal muscle function is a strong predictor for morbidity and mortality in PAD patients regardless of symptomatic presentation [7,8,9,10,11,60,61,62]. Further to this, the ischemic conditions within the limb impart a tremendous metabolic burden to the resident skeletal muscle; however little information is available about the metabolome of these tissues. Thus, the major objective of this work was to investigate the muscle metabolomics profile of CLTI patients both before surgical intervention and at amputation. These analyses uncovered a unique metabolomics profile at the time of amputation that was clearly distinguishable from Pre-surgery CLTI patients and non-PAD controls.

Accumulation of succinate in the ischemic tissues has been identified as a key regulator of reperfusion injury [63,64]. Consistent with other tissues, we observed accumulation of both succinate and fumarate in CLTI Amputation muscle (Figure 4). Interestingly, these metabolites were not different between CLTI Pre-surgery specimens and non-PAD despite clear hemodynamic evidence of limb ischemia in the former. It was recently reported that the accumulation of succinate during ischemia was primarily derived from canonical tricarboxylic acid cycle activity within minor contributions aminotransferase anaplerosis [59]. Consistent with this model, CLTI Amputation muscle displayed elevated pyruvate, alanine, glutamine, and glycine levels. This accumulation of succinate was suggested to provide a mechanism to maintain the cellular energy charge (ATP/ADP) via substrate level phosphorylation by succinyl-CoA synthetase (succinyl-CoA + ADP -> succinate + ATP + CoA). This model aligns with the metabolic challenges occurring in the ischemic skeletal muscle of failing CLTI limbs.

Striated skeletal muscles are the largest organ in human body and represent the largest mass within the ischemic limb. Skeletal muscles are a major source of branched-chain amino acid (BCAA; isoleucine, leucine, valine) catabolism where these amino acids can support tricarboxylic acid cycle flux for ATP generation. There is a growing body of evidence suggesting that elevated BCAA is a clear biomarker of cardiometabolic diseases and may play direct roles on disease pathogenesis [65,66,67]. In this study, we observed significant increases muscle BCAA concentrations in CLTI patient muscle at the time of amputation (Figure 4 and Figure 5). Considering that BCAA catabolism primarily occurs within the mitochondria, this observation aligns with previous work that identified severe deficits in mitochondrial metabolism in CLTI Amputation muscles [14]. Consistent with this notion, upon interrogation of the previously published RNA-seq data [14], CLTI Amputation muscle specimens displayed a 40–60% decrease in the expression of branched chain keto acid dehydrogenase genes (BCKDHA and BCKDHB). From a therapeutic viewpoint, it is noteworthy that defects in BCAA contribute to cardiac dysfunction [66] and stimulating BCAA oxidation in heart failure has been shown to improve cardiac function [67]. Future work is needed to determine whether stimulation of BCAA oxidation in PAD/CLTI skeletal muscle will translate to improved skeletal muscle function.

An interesting discovery from these analyses was the observation that CLTI muscle, especially those collected prior to surgical intervention, had pronounced accumulation of numerous lipid species suggesting the presence of myosteatosis. Both Pre-surgery and Amputation muscles from CLTI patients were found to have highly elevated cholesterol levels, despite the fact that all of these patients were taking statins. Cholesterol is an essential component of membranes and plays critical roles in regulating membrane fluidity and thickness. Although less abundant than the sarcolemma, mitochondrial membranes also contain cholesterol where elevated levels have been suggested to increase membrane microviscosity [68,69,70], which could play a role in the etiology of impaired mitochondrial function observed in PAD. Future work is needed to determine the subcellular location of elevated cholesterol in CLTI muscle. CLTI muscle also displayed elevated triglyceride levels, with the highest levels being detected in Pre-surgery specimens. Myosteatosis is related to increased frailty and decreased muscle function [71,72,73], both clinical phenotypes commonly observed in the CLTI population. These adverse muscle impact of myosteatosis (increase fat and decrease muscle density) was recently shown to be associated with higher mortality in PAD patients [9]. The prominent decrease in skeletal muscle mitochondrial function in CLTI muscle [14] could be related to this ectopic fat deposition; however, future work is needed to establish a causal link. Interestingly, Pre-surgery CLTI muscle specimens displayed higher levels of triglycerides compared with CLTI Amputation and control muscles. The underlying mechanism is not clear at the moment, but the observed differences suggest that the balance between triglyceride use and synthesis is far more disturbed prior to surgery than at the time of amputation.

### Study Limitations

There are several limitations to the present work that are worthy of discussion. First, the current study was cross sectional in nature, and therefore no causal inferences can be made regarding the role of metabolite differences in CLTI patients undergoing limb amputation. Future studies involving repeated muscle biopsies (e.g., pre- vs. post-surgery) and metabolite analyses are needed to determine any potential causative role in CLTI pathology leading to limb amputation. Second, this exploratory metabolomics analysis was performed on a relatively small sample size (*n* = 10 per group) and extrapolating the results to the larger population of PAD patients is cautioned. Third, there were some differences in the incidence of co-morbid conditions, principally diabetes and hyperlipidemia, as well as corresponding medication use (i.e., statins), and physical characteristics (age and sex) between CLTI patients and control participants. Matching co-morbid conditions in CLTI patients is extremely difficult, as the prevalence of many chronic diseases is substantially higher in this sub-population of PAD patients. Larger cohort studies would facilitate regression analyses to test and control for the independent effects of these factors. While NMR typically has higher reproducibility compared to MS-based metabolomics, a limitation of NMR is its lower sensitivity. Lower concentrated metabolites (i.e., nano- and pico-molar range) are not detectable using current NMR technologies. Thus, there are a substantial number of metabolites within the limb muscle that we could not detect in the current study. As these technologies improve, additional work should be performed to refine our understanding of the limb muscle metabolome in PAD/CLTI.

## 5. Conclusions

In this study, we performed a cross sectional analysis of the skeletal muscle metabolite differences in CLTI patients and non-PAD controls. We report extensive metabolite differences in CLTI muscle at the time of amputation including aberrant amino acid metabolism, myosteatosis, and classical features of ischemic/hypoxia cell metabolism. In contrast, CLTI muscles obtained from patients at the time of surgical intervention displayed relatively normal levels of amino acids with the presence of myosteatosis. These data align with the growing body of evidence that CLTI imparts a profound metabolic challenge to limb musculature and further highlights an untapped arena for therapeutic interventions aimed to improve limb metabolism and increase limb salvage. Future studies are needed to evaluate if metabolic alterations play a causal role in the limb outcomes in CLTI, including the role risk factors and co-morbid conditions.

## Figures and Tables

**Figure 1 jcm-10-00548-f001:**
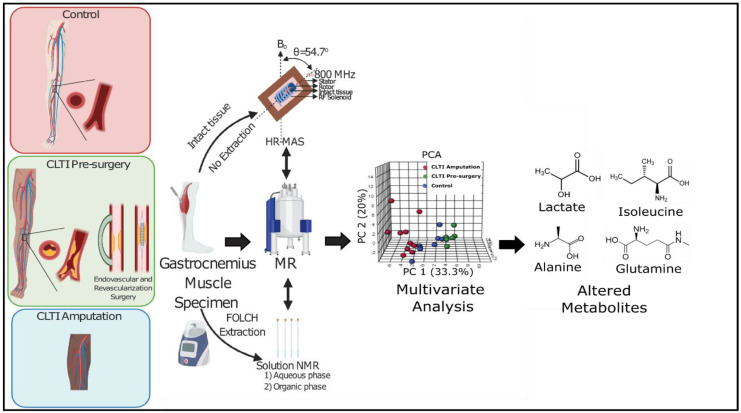
Graphical image showing the comprehensive workflow applied in this study, including gastrocnemius tissue collection from three different groups of patients (CLTI Amputation, CLTI Pre-surgery and healthy control), HR-MAS and solution state NMR spectra acquisition on those samples, multivariate analysis and identification of potential biomarkers. MR, magnetic resonance; and PCA, principle component analysis.

**Figure 2 jcm-10-00548-f002:**
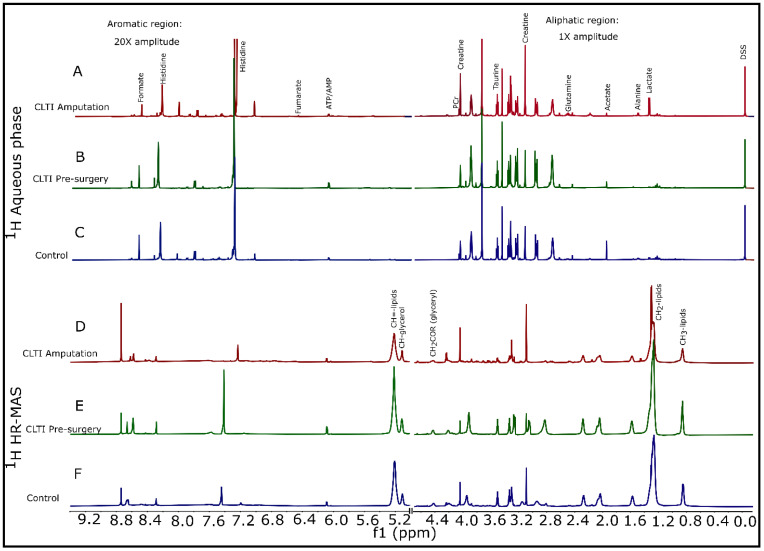
Representative ^1^H NMR spectra for aqueous phase samples (top three; normalized w.r.t. internal standard DSS peak at 0.0 ppm) and ^1^H HR-MAS (bottom three; without normalization) for gastrocnemius tissues for all three groups: control, CLTI Pre-surgery, and CLTI Amputation. Selected labeled metabolites/compounds that are significantly different between three groups are assigned for convenience.

**Figure 3 jcm-10-00548-f003:**
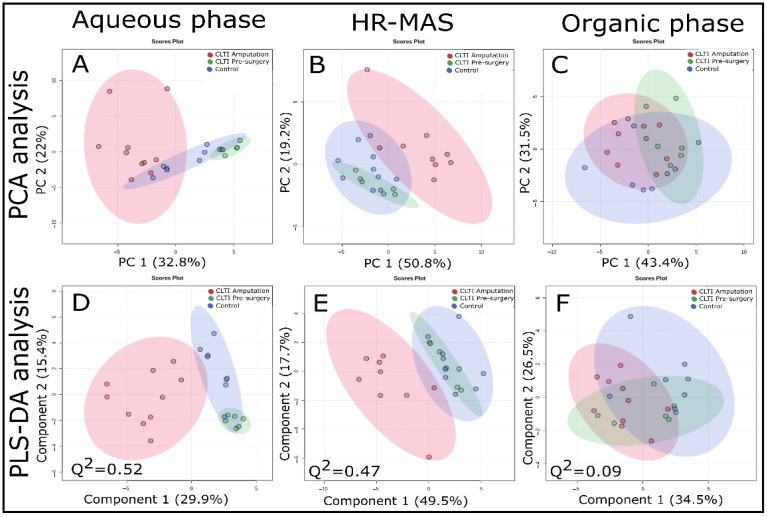
PCA and PLS-DA analyses data obtained for three different groups of human gastrocnemius samples using solution state ^1^H NMR and HR-MAS spectroscopy. Top row (**A**–**C**) were obtained from PCA analysis and bottom row (**D**–**F**) were from PLS-DA analysis. (**A**,**D**) are for aqueous phase samples (from FOLCH extraction) for PCA and PLS-DA analyses, respectively; similarly, (**B**,**E**) are for HR-MAS on intact tissue for PCA and PLS-DA analyses, respectively; and (**C**,**F**) are for organic phase (from FOLCH extraction) for PCA and PLS-DA analyses, respectively. CLTI Amputation (red), CLTI Pre-surgery (green), and control (blue).

**Figure 4 jcm-10-00548-f004:**
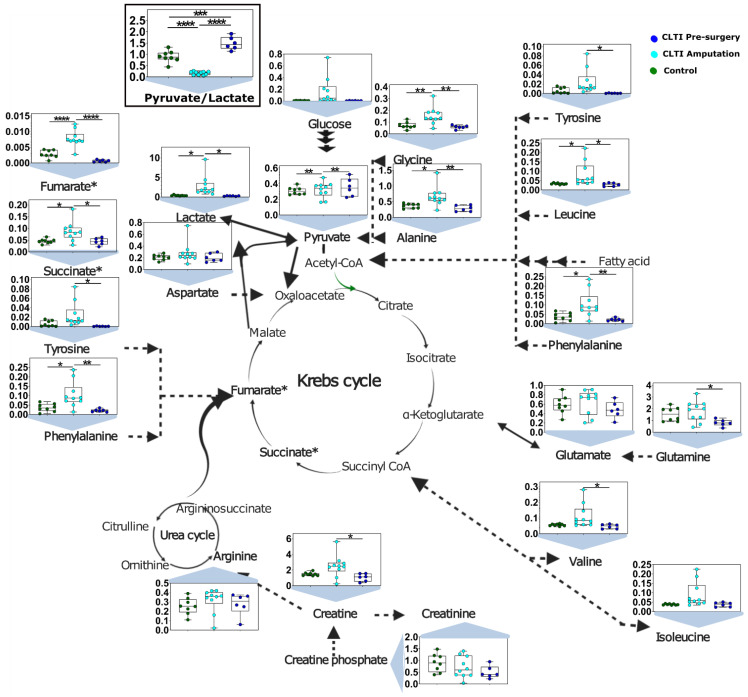
Metabolite Changes in the ischemic CLTI limb muscle. Graphical depiction and quantitative analysis of selected metabolite concentrations in muscle specimens. Box and whisker plots with 95% confidence intervals are presented for quantified amino acids with concentrations in mM. Analysis was performed using one-way ANOVA with Tukey’s multiple comparisons test. * *p* < 0.05, ** *p* < 0.01, *** *p* < 0.001, **** *p* < 0.00001.

**Figure 5 jcm-10-00548-f005:**
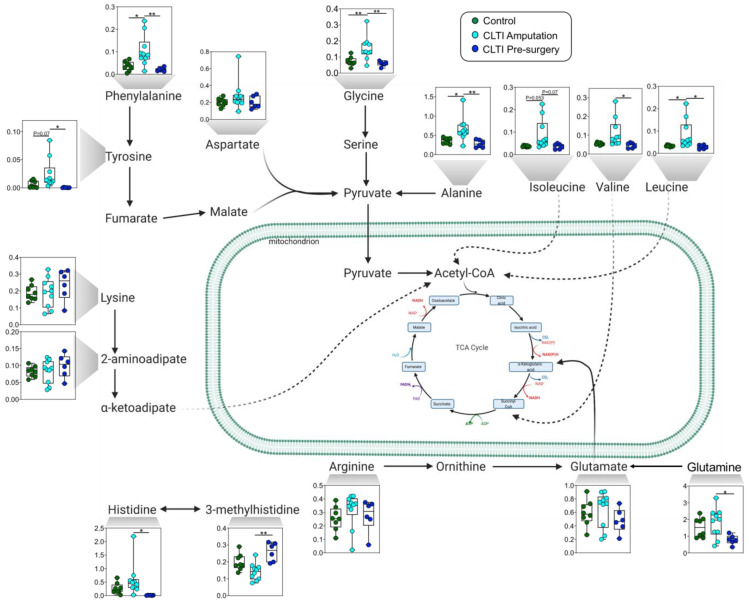
Dysregulated amino acid metabolism. Graphical depiction and quantitative analysis of amino acid concentrations in muscle specimens. Box and whisker plots with 95% confidence intervals are presented for quantified amino acids with concentrations in mM. Analysis was performed using one-way ANOVA with Tukey’s multiple comparisons test. * *p* < 0.05, ** *p* < 0.01.

**Figure 6 jcm-10-00548-f006:**
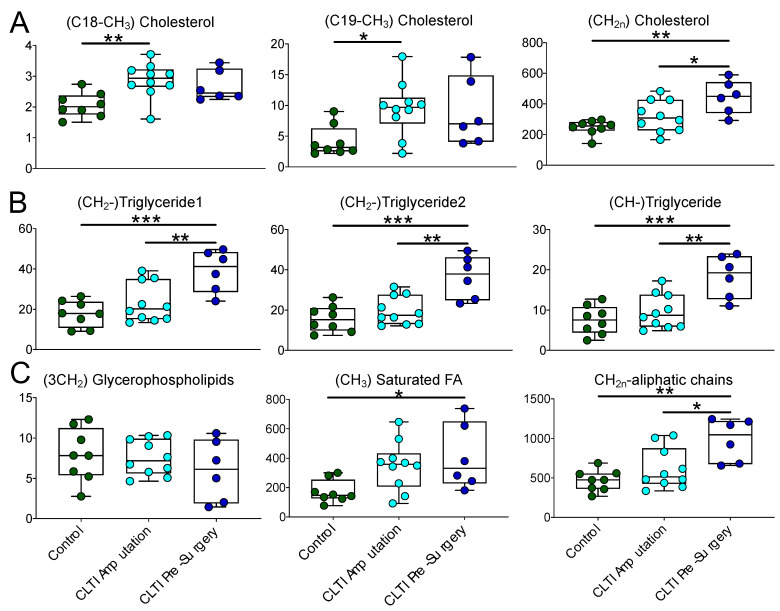
Aberrant Lipid Accumulation in CLTI muscle. Graphical depiction and quantitative analysis of lipid species in muscle specimens. Box and whisker plots with 95% confidence intervals are presented for quantified lipids are presented as peak areas. Panel **A** showed elevated cholesterols in CLTI-groups, panel **B** shows triglyceride concentrations, and panel **C** displays saturated fatty acids abundance along with similarity in glycerophospholipids amount in three groups. Analysis was performed using one-way ANOVA with Tukey’s multiple comparisons test. * *p* < 0.05, ** *p* < 0.01, *** *p* < 0.001.

**Table 1 jcm-10-00548-t001:** Patient characteristics.

		Critical Limb-Threatening Ischemia (CLTI)		
Characteristic	Control (*n* = 10)	Pre-Surgery (*n* = 10)	Amputation (*n* = 10)	*p*-Value (X^2^ or ANOVA)
Mean age (SD)—yr	73.9 (7.8)	64.5 (9.4)	69.5 (6.2)	0.043 ^A^
Female sex—*n* (%)	4 (40)	0 (0)	1 (10)	0.044
Overweight/Obese (BMI ≥ 25)—*n* (%)	9 (90)	7 (70)	8 (80)	0.535
Ankle-brachial index (ABI)—(SD)	1.1 (0.1)	0.7 (0.3)	0.5 (0.3) *	0.014 ^A^
Rutherford Classification—*n* (%)				
0	10 (100)	0 (0)	0 (0)	<0.001
3	0 (0)	4 (40)	0 (0)	0.093
4	0 (0)	2 (20)	4 (40)	0.624
5	0 (0)	4 (40)	4 (40)	0.646
6	0 (0)	0 (0)	2(20)	0.454
Medical history—*n* (%)				
Diabetes mellitus type I or II	4 (40)	6 (60)	9 (90)	0.065
Hypertension	7 (70)	10 (100)	10 (100)	0.536
Hyperlipidemia	4 (40)	10 (100)	10 (100)	0.006
Coronary artery disease	1 (10)	6 (60)	9 (90)	0.001
Renal disease	0 (0)	1 (10)	3 (30)	0.133
Chronic obstructive pulmonary disease	1 (10)	4 (40)	3 (30)	0.303
Tobacco use—*n* (%)	4 (40)	7 (70)	9 (90)	0.058
Former smoker	3 (30)	4 (40)	7 (70)	0.175
Current smoker	1 (10)	3 (30)	2 (20)	0.535
Medication used—*n* (%)				
Aspirin	4 (40)	8 (80)	9 (90)	0.035
Statin	4 (40)	10 (100)	10 (100)	<0.001
Angiotensin-converting enzyme (ACE) inhibitor	5 (50)	5 (50)	6 (60)	0.874
Cilostazol	0 (0)	3 (30)	4 (40)	0.089
Previous vascular intervention—*n* (%)	0 (0)	0 (0)	5 (50)	0.003

^A^ ANOVA was performed. X^2^ analysis was performed to determine differences in population proportions (Rutherford classifications 3–6 were only statistically compared between CLTI populations). SD, standard deviation; BMI, body mass index; ABI, ankle-brachial index; and ACE, angiotensin-converting enzyme. * Three amputation patients had non-compressible vessels precluding ABI measurement.

**Table 2 jcm-10-00548-t002:** A variable important in projection (VIP) scores obtained from partial least square discriminant analysis (PLS-DA) for metabolites from aqueous phase and HR-MAS analyses.

Spectra Range (ppm)	Metabolite	Peak Pattern	VIP Scores	
			Solution NMR (Aqueous Phase)	HR-MAS
1.31–1.33	Lactate	d	~1.9	~1.5
3.55	Glycine	s	~1.8	N.A.
2.39–2.40	Succinate	s	~1.7	N.A.
8.32–8.35	Inosine	s	~1.6	N.A.
4.04–4.06	O-Phosphoethanolamine	t	~1.6	N.A.
3.03–3.04	Creatinine + PCr	s	~1.5	N.A.
1.46–1.48	Alanine	d	~1.2	N.A.
2.32–2.36	Glutamate	m	~1.2	N.A.
3.70–3.71	3-methyl histidine	s	~1.2	N.A.
3.10	Malonate	s	~1.1	N.A.
6.88–6.90	Tyrosine	d	~1.1	N.A.
7.40–7.44	Phenylalanine	t	~1.1	N.A.
2.37–2.38	Pyruvate	s	~1.1	N.A.
2.22–2.25	2-Aminoadipate	s	~1.1	N.A.
0.84–0.95	CH_3_-lipids	m	N.A.	~1.4
5.19–5.26	CH-glycerol	m	N.A.	~1.4
1.14–1.43	(CH_2_)_n_ lipids	m	N.A.	~1.4
4.25–4.34	CH_2_OCOR (glyceryl)	dd	N.A.	~1.4
1.52–1.64	(CH_2_–CH_2_–CO–) lipids	s	N.A.	~1.3
1.93–2.11	(CH=CH–CH_2_–CH_2_) lipids	m	N.A.	~1.3
5.26–5.39	–CH= lipids	m	N.A.	~1.3
0.94–0.96	Leucine	t	N.A.	~1.1
2.70–2.90	HC=CH–C**H_2_**–HC=CH	m	N.A.	~1.1

VIP, variable important in projection; NMR, nuclear magnetic resonance; HR-MAS, high resolution magic angle spinning; s, singlet; d, doublet; dd, doublet of doublet; t, triplet; m, multiplet; and N.A., not applied. Bold font proton/s indicate the proton/s that is/are giving NMR peak/s at that particular spectral range (ppm).

## Data Availability

All raw metabolomics data have been deposited to the Metabolomics Workbench (https://www.metabolomicsworkbench.org) under the following Study ID’s: ST001615, ST001616, ST001617 and in the Appendix A.

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
