# Peer review of "Unique Metabolomic Profile of Skeletal Muscle in Chronic Limb Threatening Ischemia"

_jcm, 2021, doi:10.3390/jcm10030548_

Round 1

Reviewer 1 Report

This is an interesting analysis on CLTI patients. Here are my questions:

1)as this is not an age matched study, will age be a confounding factor in final result or analysis.

2)The same question as #1, but relates to gender.

3)The authors need to discuss more in details about whether the underlying implication is to use metabolomics as a factor to determine treatment options: amputation vs surgical/endovascular means

Author Response

This is an interesting analysis on CLTI patients. Here are my questions:

1)as this is not an age matched study, will age be a confounding factor in final result or analysis.

We thank the reviewer for this question regarding age. It is well known that the prevalence of PAD/CLTI increases with age. In thus study, our Non-PAD control group was slightly older (~74y) compared with the pre-surgery CLTI group. We did not match patients on age as this is quite difficult with rolling enrollment from our vascular clinic.  It is becoming increasing clear that PAD and CLTI manifestation of system atherosclerosis is occurring at younger ages – likely due to a multitude of factors including poor lifestyle choices (diet, exercise, smoking, etc.).  This is clear inside the clinic where is not uncommon to encounter patients in their late 40’s and early 50’s presenting with severe PAD. Nonetheless, we agree that accounting for impact of age as well as co-morbid conditions will be key to elucidating the true clinical impact – this is the focus of our efforts as we continue to generate these specimens with the goal of producing larger sample sizes for appropriate multivariate analyses that are sufficiently powered for make strong clinical claims.

2)The same question as #1, but relates to gender. We acknowledge that this study had a limited number of female participants (5 out of 30). 

We are continuing to expand our data bank and muscle biopsy biobank with the goal of performing more advanced statistical analyses as our sample size becomes large enough. Nonetheless, within the non-PAD control group, comparisons of sex did not yield statistically significant differences in any of the measures performed in this study.

3)The authors need to discuss more in details about whether the underlying implication is to use metabolomics as a factor to determine treatment options: amputation vs surgical/endovascular means

This is an excellent point raised.  We agree that the underlying metabolic complications may have implications for CLTI treatment and outcomes from treatment.  We are hesitant to make strong claims regarding the underlying implications for using metabolomics to treat or manage CLTI. Our hypothesis is that the metabolic complications the distinguish amputation from pre-surgery or even milder claudicating PAD patients (previously published by our group) may play a key role in the failure of the limb tissue to recover.  However, we have not produced a dataset yet that definitively supports this hypothesis, but we are continuing to build larger datasets which will facilitate drawing stronger conclusions.

Reviewer 2 Report

It gave me great pleasure to review the paper by Khattri and his research group. In this paper, the authors studied the metabolic profile within muscle biopsies obtained from patients with CLTI and non-pad controls.

Major comments:

  • This study had a limited sample size, “Page 3 lines 103-114” making it challenging to draw any major conclusions. Therefore, I would recommend that the authors consider making this study a pilot discovery study, and restructure the paper accordingly.
  • In the methodology section, page 3 lines 104-115, the study criteria [inclusion and exclusion criteria] for the pre-surgery, CLTI amputation as well as the none-pad control group needs to be better defined as it is not clear on how patients were selected, study time period, location on were the study was conducted (e.g ED, clinic, wards) etc...
  • There should be a dedicated specimen collection section within the methodology [lines 110–115]. As the sample preparation process is not well defined.
  • The statistical analysis section would benefit from further revision. For example, were the samples normally distributed and if so where the appropriate statistical measures taken while analyzing in none normal distributed samples.  This section will also benefit from regression analysis and there are numerous confounding factors that could have an influence on the observed findings.  Once again the low sample size number makes it challenging to conduct in-depth surgical analysis.
  • In the results section table 1, I have noticed a huge standard deviation for the ABI within the control group (SD of 0.7), as you are aware, the range for a normal ABI is between 1.29–0.91. Therefore, having a standard deviation of 0.7 with an average of 1.2 is unusual. Did the authors recruit asymptomatic PAD, patients with non-compressible ABI? Once again, the study criteria isn’t clear on this.  I anticipate that this could have risen if patients with abnormal ABI were included in the non-pad group, in such a case the TBI should have been measured, can you please elaborate more?
  • Reviewing figures 4 and 5, I have noticed that the levels of some of the identified metabolites are not consistent while comparing the control group to the amputation group or the surgery group. I feel that the results can be better presented if the authors focus on two groups g “control versus CLTI (which contains both amputation and presurgery group)?  It is not clear to me if the authors are trying to show an association between the levels of the metabolite and the severity of the CLTI (amputation/surgery group). 
  • Since regression analysis were not performed it is difficult to interpret some of the data. As you can see in table 1, most of the CLT I patients were on statins and this was statistically significant between control vs ( amputation/surgery). Does the use of statin affect the lipidomic profile in CLTI muscle?
  • In figure 6, relative to controls, some of the metabolites are significantly increased in the presurgery group and not amputation group while both of these groups are considered to have CLTI, however the authors interpretation of this finding is not explained in this paper for example triglyceride 1, triglyceride 2,  triglyceride etc..

Can you please elaborate more on this in the discussion?  Once again this could be improved if the data is presented differently by combining all CLTI patients as one cohort (amputation and presurgery ) and comparing this cohort to the controls .

Minor changes

  • The introduction is well written however it is long and it could be more concise.
  • In the introduction, lines 40-41, major reason of death in CLTI patients is multi-factorial but mainly driven by the significant atherosclerotic burden within other vascular beds such as the coronary and carotid vessels.
  • Are there any muscle biopsies obtained from patients post surgery? It will be interesting to learn the changes within the metabolic profile post arterial revascularization.
  • Please assess reformatting of the paper ensure that it follows the guidelines for this journal especially the references for example reference #47.

Author Response

It gave me great pleasure to review the paper by Khattri and his research group. In this paper, the authors studied the metabolic profile within muscle biopsies obtained from patients with CLTI and non-pad controls.

We thank the reviewer for a thorough and helpful review of this work. Our responses to each query are below in red.

Major comments:

  • This study had a limited sample size, “Page 3 lines 103-114” making it challenging to draw any major conclusions. Therefore, I would recommend that the authors consider making this study a pilot discovery study, and restructure the paper accordingly.

We agree our study has limited sample size and fully acknowledge this in the limitation sections of the discussion. This clearly limited the strength of conclusions that can be drawn, but we are continuing to collect limb specimens (typically 2-4 per week).  Our goal is to generate a larger more robust databank that will facilitate stronger statistical analysis and conclusions. We have indicated in the introduction the discovery nature of this work.

  • In the methodology section, page 3 lines 104-115, the study criteria [inclusion and exclusion criteria] for the pre-surgery, CLTI amputation as well as the none-pad control group needs to be better defined as it is not clear on how patients were selected, study time period, location on were the study was conducted (e.g ED, clinic, wards) etc...

Thank you for this suggestion.  We have added more details regarding inclusion/exclusion criteria and study locations as suggested.  These are found in lines 104-114.

  • There should be a dedicated specimen collection section within the methodology [lines 110–115]. As the sample preparation process is not well defined.

We have provided detailed specimen collection methods in our previous published papers which are referenced. These specimens were collected with standard muscle biopsy procedures (i.e. Bergstrom needles) following local anesthesia. Because this procedure has been described in detail in our previous work, we have cited these papers herein.

  • The statistical analysis section would benefit from further revision. For example, were the samples normally distributed and if so where the appropriate statistical measures taken while analyzing in none normal distributed samples.  This section will also benefit from regression analysis and there are numerous confounding factors that could have an influence on the observed findings.  Once again the low sample size number makes it challenging to conduct in-depth surgical analysis.

All data were found to be normally distributed in this work using Shaprio-Wilk tests (this detail is added now). As such, analysis of variance (ANOVA) was performed as well as the described PCA and PLS-DA analyses. We agree that multivariate analyses would help better understand the role that other clinical/physical characteristics (or co-morbidities) impact the metabolome profiles.  As the reviewer notes, the small sample size precludes the appropriate use of multivariate analyses. We did explore these options with our Statistician who recommended to apply these analyses only in large sample sizes. As described above, we are continuing to collect and perform these types of analyses on patients and anticipate a much large dataset will be available in the next several years.

  • In the results section table 1, I have noticed a huge standard deviation for the ABI within the control group (SD of 0.7), as you are aware, the range for a normal ABI is between 1.29–0.91. Therefore, having a standard deviation of 0.7 with an average of 1.2 is unusual. Did the authors recruit asymptomatic PAD, patients with non-compressible ABI? Once again, the study criteria isn’t clear on this.  I anticipate that this could have risen if patients with abnormal ABI were included in the non-pad group, in such a case the TBI should have been measured, can you please elaborate more?

We are very thankful for this question.  We have reviewed our original data files and have discovered a small mistake in the calculation performed by our study coordinator.  This mistake involved placing an incorrect selection of cells within Microsoft Excel that resulting in a single patient having an elevated ABI by calculation (miscalculated to be 2.1 which we know is physiologically unplausible).  The complete data from the control group is found below and the manuscript and table have been corrected:

Control

ABI

1.04

0.93

1.11

1.02

1.10

1.06

1.21

1.16

1.18

0.98

MEAN

1.08

SD

0.09

  • Reviewing figures 4 and 5, I have noticed that the levels of some of the identified metabolites are not consistent while comparing the control group to the amputation group or the surgery group. I feel that the results can be better presented if the authors focus on two groups g “control versus CLTI (which contains both amputation and presurgery group)?  It is not clear to me if the authors are trying to show an association between the levels of the metabolite and the severity of the CLTI (amputation/surgery group). 

We agree with the reviewers that there are statistically significant differences among groups for some metabolites. We respectfully disagree that combining CLTI pre-surgery and at amputation specimens would be a better approach. The failure rate of surgical interventions in CLTI remains high despite technically sound improvements in limb hemodynamics. There are clearly many reasons for the high failure rates, but one of our goals is to examine how the limb tissues change with interventions and whether or not the non-salvageable limbs have a set of unique characteristics which may potential allow therapeutic targeting. We hypothesize that local metabolic failure is a key component to the failed regeneration of the limb tissue following CLTI surgical interventions that ultimately result in amputation. Based on this study design, we performed ANOVA with appropriate post-hoc pairwise analyses when deemed necessary. Because of this design, we feel the appropriate presentation includes all groups as simple two-group t-testing is not appropriate for the study design.

  • Since regression analysis were not performed it is difficult to interpret some of the data. As you can see in table 1, most of the CLT I patients were on statins and this was statistically significant between control vs (amputation/surgery). Does the use of statin affect the lipidomic profile in CLTI muscle?

This is a great question and one in which we hope to address as we build our databank to a large size. The high use of statins is a fairly typically treatment for PAD/CLTI as a frontline therapy for these patients given its success in reducing overall cardiovascular mortality. Nonetheless, the data suggest that the results presented are not driven entirely by statins. Both CLTI groups were all taking statin medication, whereas the control group had 40% on statins. Despite this, in most cases the lipid levels are greater in the CLTI patients and is some cases (i.e. triglycerides) there is a significant difference between pre-surgery and amputation CLTI groups which were all taking statin medication.

  • In figure 6, relative to controls, some of the metabolites are significantly increased in the presurgery group and not amputation group while both of these groups are considered to have CLTI, however the authors interpretation of this finding is not explained in this paper for example triglyceride 1, triglyceride 2,  triglyceride etc..

This is a great observation and one which we discussed in depth during analysis of these samples. Our metabolomics analyses herein do not provide a mechanistic understanding of the biochemical causes, so at the moment we cannot be sure of the exact mechanisms. Triglycerides in the muscle specimens could be either extramyocellular or intramyocellular as whole muscle extracts were used in this study. Triglyceride levels in these samples are a result of the delicate balance between synthesis and utilization. Clear both CLI groups seem to have imbalances in these lipid portioning, although it is more pronounced prior to surgery. Future biochemical analysis of lipid synthesis and degradation pathways are necessary to determine the answer to this question. Based on this suggestion, we have add some discussion on this topic (begins at line 417).

Can you please elaborate more on this in the discussion?  Once again this could be improved if the data is presented differently by combining all CLTI patients as one cohort (amputation and presurgery ) and comparing this cohort to the controls .

We understand the reviewers request to combine CLTI cohorts. However, we firmly disagree with this suggestion. We continue to observe remarkable metabolic differences between CLTI patients prior to surgery and at the time of amputation. Because of this, we feel that combining these cohorts is inappropriate. Despite high quality surgical methods, the failure rate of endovascular and revascularization procedures in CLTI remains very high (25-40%). The reasons for these failures are undoubtedly multifactorial, but one of our major goals is to understand the metabolic changes that occur within the limb tissues pre- and post-surgery.  Thus, we are designing studies with the goal of understanding the metabolic adaptations that may contribute to limb salvage vs. amputation and feel that combining all patients would hide these distinguishing features discovered in this discovery dataset.

Minor changes

  • The introduction is well written however it is long and it could be more concise.

We have reduced the introduction wherever possible.

  • In the introduction, lines 40-41, major reason of death in CLTI patients is multi-factorial but mainly driven by the significant atherosclerotic burden within other vascular beds such as the coronary and carotid vessels.

We whole-heartedly agree with this statement and have clarified this in the introduction.

  • Are there any muscle biopsies obtained from patients post surgery? It will be interesting to learn the changes within the metabolic profile post arterial revascularization.

This is an ongoing goal of our collaborative efforts. At this moment, we have not generated enough pre/post sample collection to perform thorough analysis.  We are also continuing to collect and analyze paired pre-surgery and amputation samples for the same patients that undergo both procedures in our clinics.  However, these datasets will take many years to produce and curate the data.

  • Please assess reformatting of the paper ensure that it follows the guidelines for this journal especially the references for example reference #47.

We have revised accordingly.

Round 2

Reviewer 1 Report

Issues addressed appropriately.

Reviewer 2 Report

I would like to thank the authors for addressing my comments and reviewing the manuscript revising the manuscript. I have no additional comments.